# Nanosheets Based Approach to Elevate the Proliferative and Differentiation Efficacy of Human Wharton’s Jelly Mesenchymal Stem Cells

**DOI:** 10.3390/ijms23105816

**Published:** 2022-05-22

**Authors:** Suraj Kumar Singh, Anshuman Singh, Vinod Kumar, Jalaj Gupta, Sima Umrao, Manoj Kumar, Devojit Kumar Sarma, Marcis Leja, Manohar Prasad Bhandari, Vinod Verma

**Affiliations:** 1Stem Cell Research Centre, Department of Hematology, Sanjay Gandhi Post-Graduate Institute of Medical Sciences, Lucknow 226014, India; suraj7641@gmail.com (S.K.S.); anshumans@sgpgi.ac.in (A.S.); mail2vinod2@gmail.com (V.K.); jgupta@sgpgi.ac.in (J.G.); 2Indian Institute of Science (IISC), Bangalore 560012, India; simaphy@gmail.com; 3ICMR-National Institute for Research in Environmental Health, Bhopal 462030, India; manoj15ndri@gmail.com (M.K.); devojit.sarma@icmr.gov.in (D.K.S.); 4Institute of Clinical and Preventive Medicine, University of Latvia, LV-1586 Riga, Latvia; marcis.leja@lu.lv; 5Faculty of Medicine, University of Latvia, LV-1586 Riga, Latvia; 6Riga East University Hospital, LV-1038 Riga, Latvia

**Keywords:** Ti_2_CTx MXene, nanosheets, hWJ-MSCs, cell proliferation, cell viability, stemness, cell differentiation

## Abstract

Mesenchymal stem cell (MSC)-based therapy and tissue repair necessitate the use of an ideal clinical biomaterial capable of increasing cell proliferation and differentiation. Recently, MXenes 2D nanomaterials have shown remarkable potential for improving the functional properties of MSCs. In the present study, we elucidated the potential of Ti_2_CTx MXene as a biomaterial through its primary biological response to human Wharton’s Jelly MSCs (hWJ-MSCs). A Ti_2_CTx nanosheet was synthesized and thoroughly characterized using various microscopic and spectroscopic tools. Our findings suggest that Ti_2_CTx MXene nanosheet exposure does not alter the morphology of the hWJ-MSCs; however, it causes a dose-dependent (10–200 µg/mL) increase in cell proliferation, and upon using it with conditional media, it also enhanced its tri-lineage differentiation potential, which is a novel finding of our study. A two-fold increase in cell viability was also noticed at the highest tested dose of the nanosheet. The treated hWJ-MSCs showed no sign of cellular stress or toxicity. Taken together, these findings suggest that the Ti_2_CTx MXene nanosheet is capable of augmenting the proliferation and differentiation potential of the cells.

## 1. Introduction

Mesenchymal stem cells (MSCs) are self-renewing, multipotent cells being explored widely in clinical trials [1]. These cells have the ability to influence physiological conditions, including tissue repair and maintenance [2,3] and immuno-modulatory activities, which make them suitable for various therapeutic applications [4]. Nowadays, MSC therapy is used to treat graft-versus-host disease, multiple sclerosis, Crohn’s disease, amyotrophic lateral sclerosis, myocardial infarction, and acute respiratory distress syndrome [5,6,7]. To date, >300 clinical trials of such therapies have been completed in the patients with degenerative or autoimmune diseases [8]. Despite encouraging preclinical results, MSC therapies have limitations due to their low availability, poor differentiation efficiency and inconsistent characteristics, such as immuno-compatibility, stability, and heterogeneity [9,10]. Several efforts have been made to address these issues; one of these is the coating of culture plates (polystyrene) with different synthetic polymers, such as poly-d-lysine and natural biomaterials that are components of the extracellular matrix (ECM) [11,12,13]. Although the coating improves the cell culture conditions, the process is time consuming and expensive [14]. Tissue-derived ECM has been further used to support cell growth in three-dimensional (3D) orientation; however, it was not suitable in a monolayer [14]. Therefore, smart materials or strategies are urgently required to control the proliferation, maintenance, and differentiation of the MSCs. In recent years, various types of nanomaterials have been used to manipulate stem cells behavior by utilizing their novel properties, such as small size, high surface area, and good mechanical strength [15]. MXenes are novel class 2D materials consisting of transition metal carbides, nitrides, and carbo-nitrides (with graphene-like properties), gaining attention for numerous technological and biomedical applications [16,17]. Around 20 different MXenes with different structures and properties have been synthesized, and many more have been theoretically predicted [16]. In MXenes, titanium carbide (Ti_3_C_2_T_x_) is one of the stable and extensively studied nanomaterials for bio-applications [18,19,20]. The recently displayed osteogenic differentiation potential of Ti_3_C_2_Tx has laid the groundwork for future research to explore other MXenes with stem cells [21]. Many new MXenes members are yet to be tested for their potential in improving the stem cell functions: for example, Ti_2_CTx [22]. This MXene has a sandwich-like structure with numerous surface functionalities and is easily synthesized using a chemical etching process [22]. The physicochemical properties of Ti_2_CTx are similar to those of other 2D materials [22]. Unlike graphene and transitional metal dichalcogenides, the hydrophilicity of Ti_2_CTx renders it advantageous in stem cell research [22]. Nevertheless, to deduce its potential in the MSC research, the compatibility of Ti_2_CTx with cells must be investigated thoroughly.

Presently, there are no reports on the effect of Ti_2_CTx on the MSCs. Therefore, the current study is designed with the aim to elucidate the biological response of the Ti_2_CTx nanosheet to human Wharton’s Jelly (WJ)-MSCs by investigating the impact of nanosheets on the proliferation, viability, cellular stress, stemness, and differentiation potential of hWJ-MSCs. Our findings showed that Ti_2_CTx nanosheets are fully bio-compatible (no cellular stress) with hWJ-MSCs, and interestingly, it helped produce a significant increase in the proliferation and tri-lineage differentiation potential of the MSCs.

## 2. Results and Discussion

Owing to the promising potential of MXenes for various biomedical applications [19,20]_,_ initially, we screened three different Mxenes namely Ti_2_CTx, Ti_3_C_2_Tx, and VC_2_ to evaluate their biocompatibility with the hWJ-MSCs. However, our preliminary results demonstrated that Ti_2_CTx might be used as a potential biocompatible candidate (related primary data with Ti_3_C_2_Tx, VC_2_ are not shown) to enhance the differentiation and proliferation of the hWJ-MSCs. Therefore, the current study was designed to explore the concentration-dependent cellular behavior of Ti_2_CTx nanosheets on the hWJ-MSCs. Figure 1 shows the schematic representation of our study.

### 2.1. Characterization of Ti_2_CTx MXene

The Ti_2_CTx nanosheets were synthesized by selectively etching the aluminum layer (Al) from the MAX phase Ti_2_AlC with HF [23]. Figure 2a represents the scheme of nanosheet exfoliation. The electron microscopy and spectroscopy were used to characterize the structural, morphological and functional properties of the etched nanosheets. Figure 2b is the SEM micrograph of the synthesized nanosheets. The arrays of opened sheets are clearly visible in Figure 2b, indicating the successful synthesis of nanosheets [23].

The image showed a few layered Ti_2_CTx nanosheets with lateral dimensions of a few microns. Figure 2c depicts a TEM micrograph of synthesized nanosheets with lattice planes and some surface defects with structural variation. These variations are most likely caused by strain induced by functional groups and oxidation on the nanosheet surface [23]. High-Resolution TEM (HRTEM) shows the approximate interlayer distance of 1.5 nm (Figure 2d), which allowed determining the number of layers in etched Ti_2_CTx. Figure 2e portrays the atomic structure model of Ti_2_CTx which gave an insight into the arrangement of atoms in the synthesized nanosheets. Vibrational spectroscopy is an effective and non-invasive method for determining molecular fingerprints and the structure of 2D materials. The Raman spectra of bulk and synthesized nanosheets are shown in Figure 2f. The Raman spectrum of bulk (black curve) revealed three intense bands at 150, 265, and 365 cm^−1^ corresponding to Raman-active phonon vibration modes of ω1, ω2, ω3, and ω4. In etched nanosheets bands, ω1, ω2, and ω3 (blue curve) are blue-shifted, with reduced peak intensity, but the ω4 peak remains in the same position with reduced intensity. The peaks ω2 and ω3 are merged with the Si peak at 300 cm^−1^, and the broadening of peaks ω1 and ω4 is increased after blue-shifting, as shown in Figure 2f.

The introduced interlayer adsorbents and delamination by HF solution result in shifting and broadening in the Raman vibrational mode of Ti_2_CTx. Fourier Transform Infrared Spectroscopy (FTIR) was used to investigate the generation of various functional groups during the process of etching. The bands corresponding to functional groups -OH, -O-, and F- were seen in the FTIR spectra of etched nanosheets, indicating their presence over Ti_2_CTx. Thus, both microscopic and spectroscopic characterizations provide ample evidence of successful etching of nanosheets [23].

Unlike other conductive 2D nanomaterials with poor solubility, easy precipitation, and aggregation in a biological medium or polymeric solvents, Ti_2_CTx nanosheets’ hydrophilicity allows them to be homogeneously distributed in various solutions [16,17]. However, the hydrophilicity of a Ti_2_CTx nanosheet in combination with its high conductivity and mechanical strength [23] makes it an ideal material for stem cell-based therapeutic applications.

### 2.2. Biological Response to Ti_2_CTx Nanosheet

To elucidate the biological response of Ti_2_CTx nanosheet, a series of experiments were carried out with hWJ-MSCs. The dose-dependent effect of Ti_2_CTx nanosheets (10, 20, 50, 100, 200 μg/mL) on the morphology, proliferation, migration, viability, stress, stemness, and differentiation potential of hWJ-MSCs were explored. Post 24 h of treatment, the morphological changes were examined under a phase-contrast microscope. Both control and treated hWJ-MSCs showed normal spindle shape morphology (Figure 3a and Appendix A). Interestingly, we noticed that the cells treated with nanosheets exhibited a considerable enhancement in the proliferation of hWJ-MSCs in a dose-dependent manner (Appendix A), which was found to be maximum at 200 µg/mL concentration (Figure 3a). Upon counting the cells, the treated group (at 200 µg/mL) has shown a ≈2-fold increase in cell number than the control (Figure 3b), demonstrating the proliferation inducing property of nanosheets.

Next, a scratch assay was conducted to evaluate the effect of nanosheets on the cell migration of hWJ-MSCs [24], and the results are presented in Figure 4. After 24 h, the migration of the cells was observed in both the groups, but it is important to note that the nanosheets-treated cells showed significantly increased migration as compared to the control group. Herein, for the first time, we reported an exceptionally higher migration of hWJ-MSCs in the presence of 2D Ti_2_CTx nanosheets, which is yet another important finding of our study.

Owing to their clinical potential, we investigated the biological response of hWJ-MSCs to 2D Ti_2_CTx nanosheets and found a significant increase in cell proliferation. However, for the clinical usage of Ti_2_CTx nanosheets, other parameters, such as cell viability, cellular stress, stemness, and differentiation potential of treated hWJ-MSCs were also examined. In order to evaluate the nanosheet-mediated cellular toxicity, the MTT assay is considered as a gold standard for quantifying cell viability, metabolic activity, and proliferation [25]. The results showed a dose-dependent increase in cell viability (Figure 3c), and at the maximum concentration of nanosheets (200 µg/mL), a two-fold increase in cell viability was observed than that of the control. Thus, the nanosheet treatment enhanced the viability, metabolic activity, and proliferation of hWJ-MSCs [25]. However, these findings were inconsistent with those of previous studies, wherein concentration-dependent cell toxicity was reported by Ti_2_CTx MXenes using different cancer cell lines [26,27].

In addition to the MTT assay, propidium iodide (PI) staining was also used to corroborate the nanosheet-mediated cytotoxicity. PI is a nuclear binding dye that enters and binds to the nucleus of dead cells, producing red signals [28]. After treatment, PI was used to visualize and quantify the dead cells against Hoechst as a counterstain. Upon the quantification of PI signals, no significant difference was observed between both the groups (Figure 5b), which was further supported by confocal images of PI-stained control and treated groups (Figure 5a and Appendix A). Thus, the PI staining results further validate the MTT cell viability results, highlighting the non-toxic nature of the nanosheets.

Next, a DCF-DA assay was performed to investigate the nanosheet-mediated cellular stress (total reactive oxygen species (ROS)) in hWJ-MSCs [29]. It is a fluorometric microplate assay to detect the oxidation of DCF-DA into DCF (a highly fluorescent compound) in the presence of ROS [29]. Figure 6a (and Appendix A) is the representative confocal images of DCF-DA stained control and treated cells. No DCF fluorescence signals were detected in the treated group, highlighting the normal physiological behavior of the treated cells. Upon the quantification of DCF-DA signals, a non-significant difference in the fluorescence intensity was observed between both the groups (Figure 6b), thereby emphasizing the high biocompatibility of nanosheets with the hWJ-MSCs. In a previous study, there was a non-significant increase in the ROS level in somatic cell lines (MCF-10A, and HaCaT) and a significant increase in cancerous cells (A549) treated with Ti_2_C-PEG was reported [27], while in our study, we did not find any sign of oxidative stress in hWJ-MSCs against all tested concentrations of Ti_2_CTx, highlighting the non-toxic nature of nanosheets.

Furthermore, the impact of nanosheet exposure on the mitochondria activity of hWJ-MSCs was investigated. The mitochondria are multipurpose organelles with various cellular functions [30,31], such as being the primary producer of ROS, and they play a critical role in the physiological and pathophysiological processes including cancer development, immune functions, and blood glucose control [30,31,32]. Mitochondrial ROS (mtROS) of the treated cells were measured using the MitoSOX Red reagent [33]. It permeates and selectively targets the mitochondria of live cells, where it is rapidly oxidized by superoxide to form 2-hydroxymitoethidium. The representative confocal images of MitoSOX Red-stained control and treated hWJ-MSCs cells are shown in Figure 6c (Appendix A). No red signal was detected from either group, indicating no mitochondrial stress in the control and the treated cells. Moreover, upon quantitative analysis of the MitoSOX Red reagent (Figure 6d), no difference was observed between the control and treated groups, highlighting a healthy interaction between the cells and the material.

In this study, we also examined the expression of *Oct4*, *Sox2*, and *Nanog* stemness markers [34] in hWJ-MSCs treated nanosheet at highest tested concentration i.e., 200 µg/mL (at which the maximum proliferation and cell viability was observed), compared to the control group. Consequently, a negligible increase was observed in the mRNA levels of stemness markers in the treated group compared to the control group (Figure 7a), signifying that nanosheets induce cell proliferation without compromising the stemness of hWJ-MSCs. We also investigated the tri-lineage differentiation potential of hWJ-MSCs with and without the nanosheets. A significant change in the morphology of cells over the period of 21 days highlights the successful differentiation of hWJ-MSCs as depicted in Figure 7b. Furthermore, the differentiation was validated by performing the specific staining protocol. Unlike untreated control cells, treated hWJ-MSCs showed a significant increase in osteogenic, chondrogenic and adipogenic lineage differentiation (Figure 7b). Thus, the results demonstrate that the Ti_2_CTx nanosheet enhances the differentiation potential of hWJ-MSCs and could be further applied as an efficient tool in stem cell research and therapy.

Unlike previous studies, where MXenes have shown significant toxicity at high tested concentrations against various cell lines [35,36], our study reports a new biomaterial-like property of Ti_2_CTx nanosheets against hWJ-MSCs. Over decades, titanium oxide has been used in hard-tissue engineering owing to its structural stability and high biocompatibility [37], which favors the encouraging cell response of Ti_2_CT_X_ material. Therefore, we hypothesized that Ti_2_CTx possesses inherent non-toxic attributes of titanium and hence has no negative impact on hWJ-MSCs. Moreover, the titanium core of the material and its hydrophilicity, in combination with the high 2D surface area, provides adequate support for proliferation and differentiation without incurring cellular stress. Hence, our preliminary study unraveled a property of Ti_2_CTx that promotes human WJ-MSC proliferation without compromising its stemness and differentiation potential. This phenomenon is supported by the high biocompatible nature of nanosheets. The excellent biological competence of the nanosheets can be further explored to address some of the current challenges in mesenchymal and induced pluripotent stem cells differentiation into certain lineages and organoids for disease modeling. The limitation of our study is that it does not provide mechanistic insight into the nanosheet-mediated enhanced proliferation and differentiation of hWJ-MSCs. Moreover, the interaction of nanosheets with the cells needs to be examined further in the future research studies.

## 3. Experimental Design

### 3.1. Materials

Ti_2_AlC, Ti_3_AlC_2_, and V_2_AlC powders (bulk) were purchased from 3-ONE-2, Voorhees, NJ, USA. Lithium Fluoride (LiF), Hydrogen Fluoride (HF), Hydrogen Chloride (HCl), and Tri reagent (Cat No. T9424) were procured from Sigma Aldrich, Bangalore, India. α-MEM culture media (Cat No. 11900024), Antibiotics (Cat No. 15140-122), L-glutamax (Cat No. 35050-061), and Fetal Bovine Serum (FBS; Cat No. 12662029) were purchased from Gibco, Thermo Fisher Scientific, Waltham, MA, USA. Non-essential amino acids (Cat No. ACL006) and hWJ-MSCs (Cat no. CL005) were procured from Hi-Media Laboratories, Maharashtra, India.

### 3.2. Synthesis Procedure of Ti_2_CTx MXenes

Ti_2_CTx MXene was prepared following the protocol of Yang et al. 2017 [23]. In brief, 1 g of Ti_2_AlC powder was added into a freshly prepared solution of LiF (1.8 g) in 12 M HCl (30 mL), and the mixture was kept for 30 min in ice cold water to avoid overheating, and it was incubated at 37 °C for overnight using a water bath. The resulting suspension was then immersed in 10% HF solution (for 10 min) and washed ≈16 times with deionized (4000 rpm × 5 min for each cycle) until the pH of the supernatant reached ≈7. Thereafter, the suspension was sonicated for 10 min, centrifuged at 1000 rpm (for 10 min) to remove bulk Ti_2_CTx, and finally, the supernatant was centrifuged at 6000 rpm for 10 min to obtain the Ti_2_CTx as a by-product. In addition, we have also synthesized the Ti_3_C_2_Tx and VC_2_ nanosheets following standard protocols [38,39].

### 3.3. Characterization of Nanosheets

Scanning and transmission electron microscopy (SEM and TEM) were used to investigate the structural and morphological characteristics of the synthesized nanosheets. Raman and Fourier Transform Infrared Spectroscopy (FTIR) spectroscopies were used to probe the optical properties of the materials. For SEM and Raman analysis, samples were drop casted on a silicon wafer. FTIR analysis was completed using a KBr pellet saturated with material. For TEM analysis, nanosheet suspension was dropped on a Formvar coated copper grids (300 mesh), and then, images were acquired.

### 3.4. Biological Activity

Initially, we considered three MXenes, i.e., titanium carbides (Ti_2_CTx and Ti_3_C_2_Tx) and vanadium carbide (VC_2_) for biological studies with hWJ-MSCs, and on the basis of the preliminary results, we continued with Ti_2_CTx (data related to the Ti_3_C_2_Tx, VC_2_ are not shown).

#### 3.4.1. hWJ-MSCs Cell Culture

Cryopreserved passage 3 (P3) hWJ-MSCs were thawed and seeded in a T25 flask in a complete cell culture medium (α-MEM medium supplemented with 15% FBS, 1× non-essential amino acid, 1× antibiotic gentamycin–amphotericin, 1× glutamax) at 37 °C with 5% humidified CO_2_ atmosphere. Once the cells reached 80–90% confluency, they were detached using Trypl E, counted, and re-seeded in culture plate for further investigation. For colorimetric/fluorometric analysis (MTT, PI, DCF-DA and MitoSOX), cells were seeded at a density of 2 × 10^3^ in 96-well culture plate in culture media and were allowed to reach 60–70% confluency. Later on, cells were replaced with the fresh medium supplemented with Ti_2_CT_X_ nanosheets at different concentrations (10, 20, 50, 100, 200 µg/mL) for 24 h. Post-treatment, cells were washed thrice with 1X PBS and then incubated with fresh growth medium without Ti_2_CT_X_ nanosheets for the next 24 h. After 24 h, the treated and control cells were visualized in contrast microscope for any morphological aberrations and to assess their proliferation.

#### 3.4.2. Scratch Assay

Scratch assay [24] was performed in order to evaluate the effect of Ti_2_CTx in hWJ-MSCs migration. The cells were seeded at a density of 1 × 10^4^ cells per well in a 24-well plate and were allowed to attain 60–70% confluency. Post confluency, the media was replaced with fresh growth media, with and without nanosheets, and a scratch was made in the well of both the groups (Day 0), and the cells were allowed to grow for the next 24 h. After 24 h (Day 1), cell migration was analyzed using the phase contrast microscope.

#### 3.4.3. MTT Assay

The MTT [3-(4,5-dimethylthiazol-2-yl)-2,5-diphenyltetrazolium bromide], a colorimetric test [25], was used to determine the effect of Ti_2_CTx nanosheets on cell viability, metabolic activity, and proliferation of hWJ-MSCs. The treated and control cells were incubated for another 4 h at 37 °C with 5% humidified CO_2_ atmosphere with media containing MTT (0.5 mg/mL final concentration). Thereafter, a solubilizing agent (DMSO, 100 µL per well) was added to each well and was kept overnight in the dark in a humidified environment. The absorbance of the formazan produced was measured at 570 nm. Each dose was tested at least three times. According to Formula (1), the results were presented as a percentage of cell viability in comparison to the control groups [25].
(1)$$\mathrm{Cell}\;\mathrm{Viability}=\frac{\mathrm{Ai}}{\mathrm{Ac}}\times 100\%$$
where Ai = average absorbance of treated cell, Ac = average absorbance of control cells.

#### 3.4.4. Cell Viability Assessment via Staining with Propidium Iodide

Propidium iodide (PI) is a dye used to distinguish between live and dead cells [28]. The treated and control cells were incubated for 30 min with fresh PBS containing PI (10 µg/mL final concentration) and Hoechst (10 µg/mL final concentration). The PI and Hoechst were quantified using spectrophotometer with an excitation and emission wavelengths at 493 nm and 636 nm, and 352 nm and 455 nm, respectively. Moreover, both the groups of cells were also examined under confocal microscopy in a separate set of experiments with DAPI as a counter stain.

#### 3.4.5. Total ROS Production

A fluorometric microplate test capable of detecting the oxidation of 2′,7′-dichlorofluorescin-diacetate (DCF-DA) into 2′,7′-dichlorofluorescein (DCF, a highly fluorescent chemical) in the presence of reactive oxygen species was used to examine the oxidative stress generated by Ti_2_CTx nanosheets [29]. The treated and control cells were incubated for 30 min with fresh PBS containing DCF-DA (10 µg/mL final concentration) and Hoechst (10 µg/mL final concentration). Thereafter, DCF-DA fluorescence was quantified using spectrophotometer at excitation and emission wavelengths of 495 nm and 529 nm for DCF-DA and 352 nm and 455 nm for Hoechst, respectively. The confocal microscope was used to examine the DCF-DA/Hoechst stained cells in the FITC and UV channel, respectively, in a separate experiment with DAPI as a counter stain. Each dose was tested a minimum of three times. According to Formula (2), the results were presented as a fluorescence percentage of DCF-DA in comparison to the control groups [29].
(2)$$\mathrm{ROS}=\frac{\mathrm{IFi}}{\mathrm{IFc}}\times 100\%$$
where IFi = average fluorescence of treated cells, IFc = average fluorescence of control cells.

#### 3.4.6. Mitochondrial Superoxide Detection via MitoSOX-Red Indicator

Mitochondrial ROS (mtROS) are the most important and promising targets for understanding mitochondrial diseases [33]. mtROS of the treated cells were estimated using MitoSOX^TM^ Red dye [33]. The treated and control cells were incubated for 30 min with fresh PBS containing MitoSOX^TM^ Red dye (10 µg/mL final concentration) and Hoechst (10 µg/mL final concentration). MitoSOX was quantified using a spectrophotometer at excitation and emission wavelengths of 510 nm and 580 nm for mtROS and 352 nm and 455 nm for Hoechst, respectively. The confocal imaging was completed to examine MitoSOX positive cells in the TRITC channel and UV channel with DAPI as a counter stain.

#### 3.4.7. Stemness and Differentiation Potential

##### RT-PCR Analysis

Total RNA from hWJ-MSCs was extracted by TRIzol reagent. The RNA was reverse transcribed into cDNA using a High-capacity cDNA Reverse Transcription Kit (Thermo Fisher, Waltham, USA). RT-PCR was carried out using Takyon No ROX SYBR mastermix (Eurogentec; Cat no. UF-NSMT-B0701) according to the manufacturer’s protocol. The primer sequences were as follows: β-Actin forward primer 5′-CCCTGGACTTCGAGCAAGAG-3′ and reverse primer 5′-ACTCCATGCCCAGGAAGGAA, human Sox2 forward primer 5′-AAAAATCCCATCACCCACAG-3′ and reverse primer 5′-GCGGTTTTTGCGTGAGTGT-3′, human Nanog forward primer 5′-CTCCATGAACATGCAACCTG-3′ and reverse primer 5′-GAGGAAGGATTCAGCCAGTG-3′, human Oct4 forward primer 5′-AGTTTGTGACAGGGTTTTTG-3′ and reverse primer 5′-ACTTCACCTTCCCTCCAACC-3′. All primers were synthesized in Integrated DNA Technologies (IDT).

##### Tri-Lineage Differentiation of MSCs

The WJ-MSCs P3 (control and treated) were cultured in 6-well plates (5 × 10^5^ cells/well) supplemented with an osteogenic and adipogenic differentiating media kit (Thermo Fisher), respectively. The cells were incubated for 14–21 days, and the medium was changed every 3–4 days. For chondrogenic differentiation, the cells were seeded in 96-well plates (1 × 10^4^ cells/well) with a chondrogenic differentiation media kit (Thermo Fisher) for the next 14–21 days. Post incubation, the cells were washed thoroughly with 1X PBS, and differentiated cells were stained with lineage-specific dyes: Alcian Blue for chondrogenic cells, Alizarin Red S stain for osteogenic cells and Oil Red O for adipogenic cells.

#### 3.4.8. Statistical Analysis

The SPSS22.0 and Graph Pad prism 5 software were used to evaluate all of the results. The data were presented in the form of means and standard deviations (SD). One-way analysis of variance (ANOVA) was used for the statistical analysis. To compare various groups, Tukey HSD post hoc testing was utilized as the post hoc correction. Two-tailed unpaired Student’s t-tests were performed to compare the differences between the two groups. At *p* = 0.05, differences were judged statistically significant.

## 4. Conclusions

We developed an efficient and robust MXene nanosheet-based approach for the large-scale expansion and differentiation of WJ-MSCs. Moreover, a two-fold increase in cell viability was also noticed without any visible sign of cellular stress at the highest tested dose of the nanosheet. In conclusion, the Ti_2_CTx MXene nanosheet could be used as a robust platform for regenerative medicine applications, in particular for stem cell growth, proliferation, and differentiation.

## Figures and Tables

**Figure 1 ijms-23-05816-f001:**
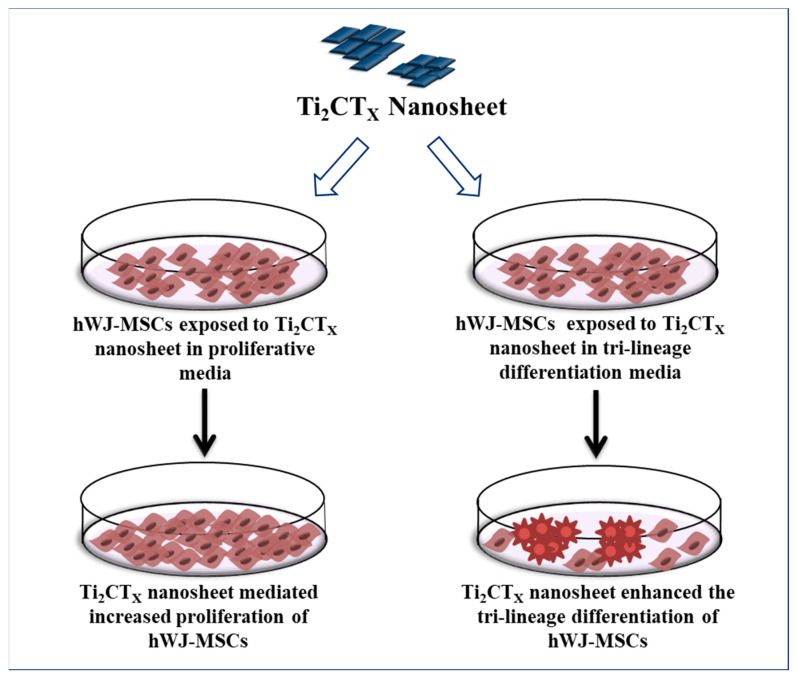
Schematic representation of research study.

**Figure 2 ijms-23-05816-f002:**
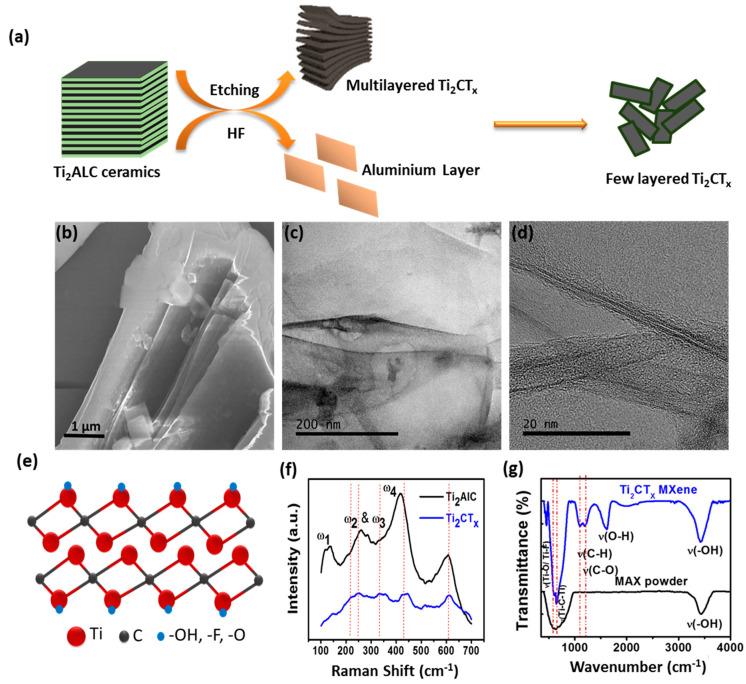
Characterization of etched Ti_2_CTx nanosheets. (**a**) Schematic showing etching of nanosheet; (**b**) SEM micrograph showing a few layered nanosheets; (**c**,**d**) TEM and HRTEM micrographs confirming the layer composition of nanosheet; (**e**) Model showing atomic structure of Ti_2_CTx; (**f**) Raman spectra confirming a successful production of nanosheet from bulk; (**g**) FTIR spectra showing the presence of various functional groups over etched Ti_2_CTx.

**Figure 3 ijms-23-05816-f003:**
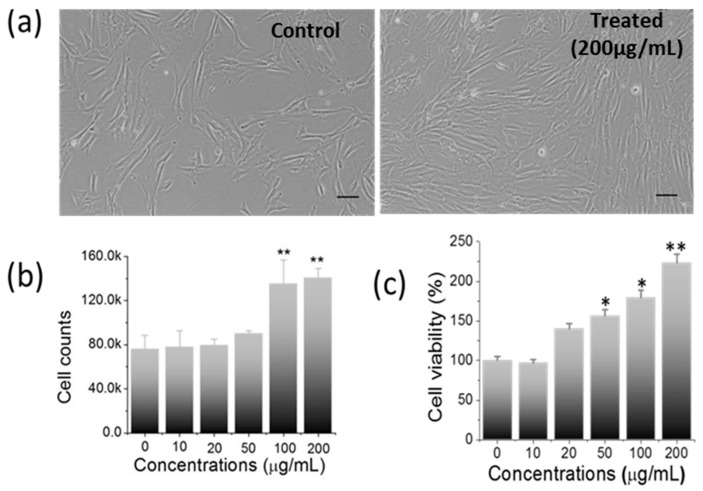
Impact of nanosheet treatment on cellular behavior. (**a**) Phase-contrast micrographs showing normal fibroblast-like morphology in treated hWJ-MSCs with an enhanced proliferation at 200 µg/mL than control (Scale bar: 50 µm), (**b**) Graph showing ≈2-fold enhanced cell count in treated cells vs. control cells, (**c**) Significant increase in cell viability analyzed by MTT assay. (* *p*-value < 0.05; ** *p*-value < 0.001).

**Figure 4 ijms-23-05816-f004:**
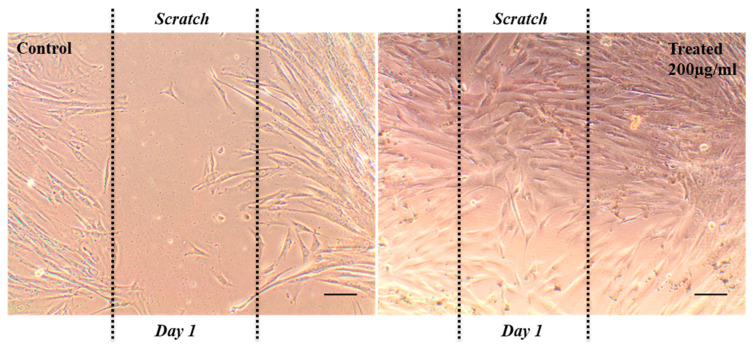
Phase-contrast images of migration assay showing the enhanced cell migration in treated cells (200 µg/mL) than control in scratch area (scale bar: 50 µm).

**Figure 5 ijms-23-05816-f005:**
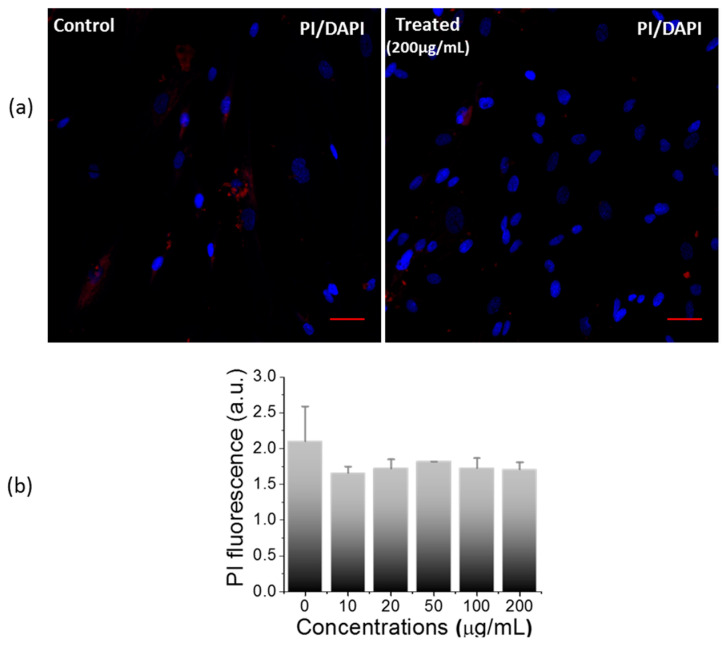
Impact of nanosheet on cell cytotoxicity. (**a**) Confocal images of PI staining for evaluation of nanosheet mediated cell cytotoxicity (scale bar: 50 µm), (**b**) Cytotoxicity assessment by quantitative estimation of PI between control and treated cells (200 µg/mL).

**Figure 6 ijms-23-05816-f006:**
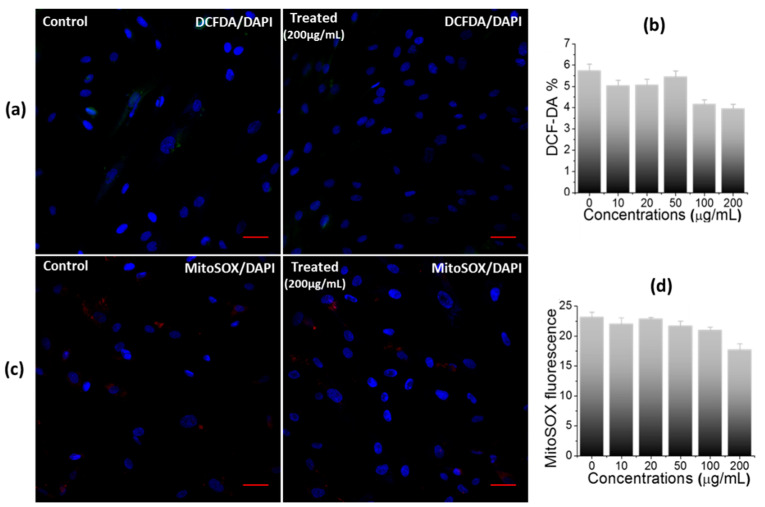
Impact of nanosheets treatment on oxidative stress in hWJ-MSCs. (**a**) Confocal images, (**b**) Quantitative estimation of DCF-DA stained control and treated cells (200 µg/mL), showing no oxidative stress, (**c**) Confocal images, and (**d**) Quantitative estimation of MitoSOX-stained control and treated cells, indicating non-significant generation of mitochondrial superoxide in treated cells (scale bar: 50 µm).

**Figure 7 ijms-23-05816-f007:**
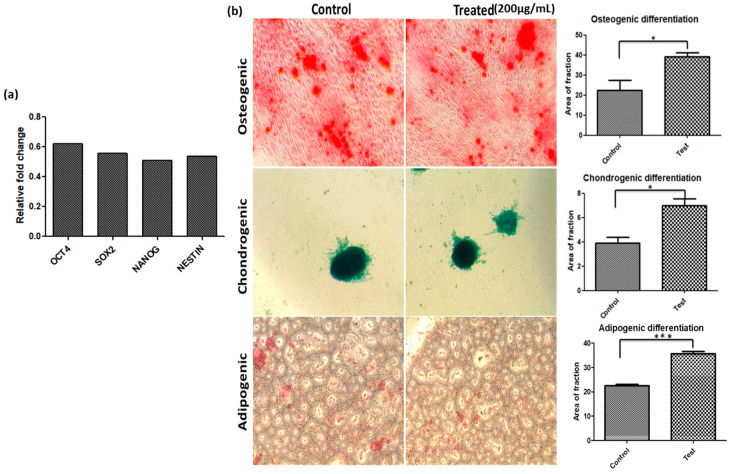
Impact of nanosheet on stemness and tri-lineage differentiation potential of WJ-MSCs. (**a**) Expression of stemness marker, (**b**) Phase contrast microscopic images and gene expression analysis showing the enhanced tri-lineage differential potential of WJ-MSCs with the nanosheet. (* *p*-value < 0.05; *** *p*-value < 0.001).

## Data Availability

Not applicable.

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
