# Peer review of "Nanosheets Based Approach to Elevate the Proliferative and Differentiation Efficacy of Human Wharton’s Jelly Mesenchymal Stem Cells"

_ijms, 2022, doi:10.3390/ijms23105816_

Round 1
Reviewer 1 Report
The authors aimed to investigate the impact of nanosheets on the roliferation, viability, cellular stress, stemness and differentiation potential of hWJ-MSCs. Their findings showed that Ti2CTx nanosheets are fully bio-compatible (no cellular stress) with hWJ-MSCs, and interestingly, it helped in significant increase in their proliferation and tri-lineage differentiation potential.
The study covers some issues that have been overlooked in other similar topics. The structure of the manuscript appears adequate and well divided in the sections. Moreover, the study is easy to follow, but some issues should be improved. Some of the comments that would improve the overall quality of the study are:
- Authors must pay attention to the technical terms acronyms they used in the text.
- English language needs to be revised.
- Limitations of the study needs to be added.
- Conclusion Section: This paragraph required a general revision to eliminate redundant sentences and to add some "take-home message".
Reviewer 2 Report
Authors have done intensive biological test with human Wharton’s Jelly MSCs (hWJ-MSCs) and Ti2CTx nanosheets. The experiment methods described in detail. The cell culture was conducted with different concentration of Ti2CTX nanosheets for 24 h. Post-treatment, cells were washed thrice with 1X PBS, and then incubated with fresh growth medium without Ti2CTX nanosheets for the next 24 h. It can be obtained that the Nanosheet is around 3 um width from figure 2b and the size of Mesenchymal stem cell is about 100 um from figure 4. There may be chance the Mesenchymal stem cell uptake the Ti2CTX nanosheets or nano sheet grooved into the cells. Have you study this phenomenal. Please address this issue.
It requests indication of scale bar for Figure 5 a.
Round 2
Reviewer 2 Report
THe manusrcript has been improved.
Figure 5
There are scale bars in figure a ,but there is description. please include scale bar size in the figure legend.
Figure S2-Figure S4
There is description scale bars sizes in the figure legend in figure S2-S4. There are not scale bars in the images, please include all of scale bars in the six panels of figure S2 , Figure S3, Figure S4.
